# Transmission Network of Deer-Borne *Mycobacterium bovis* Infection Revealed by a WGS Approach

**DOI:** 10.3390/microorganisms7120687

**Published:** 2019-12-12

**Authors:** Lorraine Michelet, Cyril Conde, Maxime Branger, Thierry Cochard, Franck Biet, Maria Laura Boschiroli

**Affiliations:** 1Paris-Est University, National Reference Laboratory for Tuberculosis, Animal Health Laboratory, French Agency for Food, Environmental and Occupational Health and Safety (Anses), 94701 Maisons-Alfort CEDEX, France; lorraine.michelet@anses.fr; 2Infectiologie et Santé Publique (ISP), Institut National de Technologie Agronomique (INRA), Université de Tours, UMR 1282, 37380 Nouzilly, France; cyril.conde@inra.fr (C.C.); maxime.branger@inra.fr (M.B.); thierry.cochard@inra.fr (T.C.); franck.biet@inra.fr (F.B.)

**Keywords:** bovine tuberculosis, *Mycobacterium bovis*, deer herd, animal park, WGS, SNP calling

## Abstract

Bovine tuberculosis (TB) is a zoonotic disease, mainly caused by *Mycobacterium bovis*. France was declared officially TB free in 2001, however, the disease persists in livestock and wildlife. Among wild animals, deer are particularly susceptible to bovine TB. Here, a whole genome sequence (WGS) analysis was performed on strains with the same genetic profile—spoligotype SB0121, Multiple Loci VNTR Analysis (MLVA) 6 4 5 3 11 2 5 7—isolated from different types of outbreaks, including from deer or cattle herds, or zoological or hunting parks where the presence of infected deer was a common trait in most of them. The results of the phylogeny based on the SNP calling shows that two sub-clusters co-exist in France, one related to deer bred to be raised as livestock, and the other to hunting parks and zoos. The persistence over almost 30 years of sporadic cases due to strains belonging to these clusters highlights the deficiency in the surveillance of captive wildlife and the need for better monitoring of animals, especially before movement between parks or herds.

## 1. Introduction

Bovine tuberculosis (bTB), mainly caused by *Mycobacterium bovis*, is an important re-emergent zoonotic disease in Europe [1]. Although France has been officially bTB free (OTF) since 2001, the persistence of the disease in livestock and its occurrence in some areas of wildlife are matters of great concern [2]. This ancient disease is no longer considered just a cattle-borne problem, but a concern for multihost communities that include wildlife species such as wild boar (*Sus scrofa*), red deer (*Cervus elaphus*), and badgers (*Meles meles*) [3].

In Europe, red and fallow deer (*Dama dama*) are frequently reported to be infected by *M. bovis* or *M. caprae* [4]. In the wild population, red deer are considered to play an important role in the dissemination of the disease as an important number of infected animals disclose generalized patterns of clinical tuberculosis [5]. Farmed deer are also very susceptible to *M. bovis*, as a high incidence of animals with generalized infection is commonly observed in deer herds [6]. Gross lesions are generally observed in lymph nodes of the head, thorax and abdomen [4]; most frequently in the retropharyngeal lymph nodes for red deer, and the thoracic region for fallow deer [7].

Interaction among different wildlife species, and between wildlife and human beings, may now be more frequent due to the increasing amount of captive wildlife in zoos or other types of animal parks [8]. The zoonotic transmission of *Mycobacterium tuberculosis* complex (MTBC) pathogens has been reported worldwide: e.g., an *M. bovis* infection in zookeepers exposed to a white rhinoceros [9], an *M. pinnipedii* infection in animal handlers linked to a sea lion colony [10], and an *M. tuberculosis* infection of staff members exposed to elephants [11,12]. Infrastructure where the public has potential contact with captive wildlife is a serious emerging concern for zoonotic transmission, especially for immunocompromised individuals [8]. The same concern arose after the discovery in an Iranian zoo of an *M. bovis* infection in free-roaming fallow deer that had been in close contact with the public [13].

MTBC members are excellent microorganisms for conducting molecular epidemiology studies on, by virtue of their low mutation rate nature. In the case of bTB, two genotyping methods are widely used—spoligotyping [14] and Multiple Loci VNTR Analysis (MLVA) typing [15]—both of which establish differentiation between *M. bovis* strains, taking into account relative genetic stability. The combination of these two techniques allows for a very fine differentiation of strains. In France, 497 different genotypes could be identified between 1978 and 2013, in a collection of *M. bovis* strains isolated from cattle and wild animals [2,16]. The vast diversity of the molecular profiles obtained by combining these two techniques makes it possible, by comparing the profiles of isolated strains of cattle and wild animals, to determine the origin of the infection of a very large number of outbreaks and to highlight possible inter-species transmission [17]. However, in areas where the disease is still highly prevalent, unique dominant genetic profiles-per-zone are shared by almost all isolates, which makes the reconstitution of a transmission chain impossible [17,18]. Molecular techniques with finer resolutions are necessary in such contexts. Analyses of mycobacterial whole genome sequences (WGS), which detect genomic changes at a very small scale, hold promise as a strain discrimination alternative [19]. This technique has particularly been used to better understand the human-to-human transmission of different mycobacteria, such as *M. leprae* [20] or *M. tuberculosis* [21]. WGS data have also been used to reconstruct transmission scenarios of *M. bovis* infection [22,23]. 

The aim of this study was to use WGS analysis in order to investigate the transmission network of *M. bovis* with a particular genotype, mainly identified in infected red deer belonging to French farms, zoos, or hunting or other type of parks. 

## 2. Material and Methods

### 2.1. Ethical Statement

In France, bTB is a notifiable disease with an eradication program in cattle and surveillance in free range wildlife. The official methods for the diagnosis of this disease are culture, PCR and histopathology. Therefore, all the samples included in this study are issued from animals analyzed within an official context. No purposeful killing of animals was performed for this study. All samplings were in complete agreement with national and European regulations. No ethical approval was necessary. 

### 2.2. Bacterial Strains

This study included 24 field *M. bovis* isolates belonging to the French bTB National Reference Laboratory (NRL, Anses) collection (Table 1). The source population was animals whose samples were analyzed because (i) they presented non-negative skin tests and/or γ-interferon tests results and were culled for diagnostic purposes; (ii) they were culled as a result of the total or partial slaughter of their herd of origin after confirmation of the herd’s infection, or (iii) they presented macroscopic bTB-like lesions at routine abattoir inspection. Bacterial culture was performed following the protocol established by the French NRL (NF U 47–104) for the isolation of *M. bovis*. The identity of the isolated *Mycobacterium tuberculosis* complex colonies was confirmed by DNA amplification as described by Hénault et al. [24], and *M. bovis* was confirmed by spoligotyping and VNTR typing as described below. For livestock breakdowns, a herd-based criterion was used for strain choice: at least 1 strain/outbreak was selected. Spoligotyping was performed as described by Zhang et al. [14], using TB-SPOL kits purchased from Beamedex^®^ (Beamedex^®^ SAS, Orsay, France) on BioPLex200/Luminex 200^®^ as described by Hauer et al. [2]. Spoligotypes have been named according to an agreed international convention (www.mbovis.org). A Multi-Locus Variable numbers tandem repeats Analysis (MLVA) profile identification [25] was performed by Genoscreen (Lille, France), using PCR amplification targeting genetic loci including mycobacterial MIRU-VNTR. Analysis was based on 8 loci, ETR A (VNTR2165), ETR B (VNTR2461), ETR C (VNTR577), ETR D (MIRU4 or VNTR580), QUB 11a (VNTR2163a), QUB 11b (VNTR2163b), QUB 26 (VNTR4052) and QUB 3232 (VNTR3232), chosen in the framework of a European consortium based on their degree of polymorphism and their ability to discriminate local strains [26].

### 2.3. Whole Genome Sequencing and Variant Calling

Whole genome paired end (2 × 250 bp) sequencing was performed on Rapid Run HiSeq 2500 (Illumina) using NextEra XT libraries (Illumina, San Diego, CA, United States), and indexed following the manufacturer’s recommendations (Illumina, San Diego, CA, United States) by Genoscreen (Lille, France). The theoretical coverage was expected to be higher than 50×. Raw sequences were aligned to the reference genome AF2122/97 (Genbank accession NC_002945.4) using BioNumerics v7.6 (Applied Maths). A wgSNP analysis was performed with the strict SNP filtering template, i.e., removing positions with at least one unreliable base, ambiguous base, or gap; removing nondiscriminatory positions; with a minimum of 10× coverage; with a minimum distance of 12 nucleotides between SNPs. The evolutionary history of this dataset was inferred on MEGA7 software [27], using the Maximum Likelihood method based on the General Time Reversible (GTR) model (model selected based on jmodeltest v2.1.10 [28]). A bootstrap analysis was performed with 500 replicates.

WGS data were submitted to the EBI/EMBL. The accession Numbers are listed in Table 1.

## 3. Results

Among the 86 strains available at the NRL with the genotype of interest, i.e., a spoligotype profile belonging to the European 2 complex (either GB54/SB0121 or F70/SB0295) and a unique VNTR profile (6 4 5 3 11 2 5 7), 23 strains were selected from different herds or other origins, to represent all potential outbreaks (Table 1). They were isolated between 1979 and 2018 from 12 cattle (*Bos taurus*), 3 bison (*Bison bonasus*), 3 wild boar (*Sus scrofa*), 3 red deer (*Cervus elaphus*), 1 fallow deer (*Dama dama*), 1 wapiti (*Cervus canadensis*) and 1 nonhuman primate (species not specified). The strain isolated in 1979 from a wild boar was selected as an outgroup, as it shared the same spoligotype but a slightly different VNTR profile (6 3 5 3 11 1 5 7).

These isolates were analyzed by WGS. In total, we identified 770 SNPs compared to the *M. bovis* AF2122/97 reference genome (Appendix A). The phylogenetic analysis highlighted different clusters, all enrooted by the wild boar strain isolated in 1979 (Figure 1 and Figure 2). A distance matrix of SNPs was generated using MEGA (Table 2) and the number of SNPs between each strain was determined. A first cluster supported by a bootstrap value of 100% is defined by 47 SNPs. This cluster is sub-divided into two sub-clusters: the first one, which is phylogenetically well supported (cluster 1.1, 98%), containing strains isolated from the cattle or deer of livestock herds; the second one, which is less robust (cluster 1.2, 67%), contains strains isolated from zoo animals or belonging to hunting parks (Figure 2). A low number of SNPs defined these sub-clusters—four and one respectively. Cluster 2 groups all the strains with spoligotype F70/SB0295, isolated from one French cow and from foreign-born animals, and is defined by 24 SNPs. Cluster 3, defined by 51 SNPs, contains strains isolated from foreign-born animals with spoligotype GB54/SB0121.

The 16 French strains gathered in cluster 1 seem to be closer (between 0 and 46 SNPs) than the strains of cluster 2 and 3. Indeed, some of them have very few genetic differences. Two strains isolated from a bison and a wapiti from the same animal park are identical. The WGS analysis allows us to confirm a link between two strains isolated from one wild boar in 2007, another in 2012 and a strain isolated from a fallow deer in 2017. The total number of SNPs for cluster 1 is 107, 50 for cluster 1.1, and 57 for cluster 1.2. On the other hand, the number of differences in the remaining clusters is higher with fewer strains, i.e., 163 SNPs for cluster 2 and 199 SNPs for cluster 3.

## 4. Discussion and Conclusions

This investigation, based on genomic data, allowed us to reveal the circulation of closely related *M. bovis* strains that seem to be confined to deer farms and animal parks. Although these types of strains have been regularly found since the 1990s, no geographical link seems to exist between most herd cases. WGS analysis showed a main cluster in France, divided into two very close sub-clusters (with just five SNP differences)—cluster 1.1 related to cattle and deer herds reared as livestock, and cluster 1.2 related to recreational or hunting animal parks. This analysis confirms the infection by the same strain of a bison and a wapiti from the same park. It also shows a link between the strain isolated recently in a fallow deer and two wild boar strains in 2007 and 2012 [29]. An epidemiological link exists between these three cases: the wild boars belonged to the same hunting park and the fallow deer, found infected but in another animal park, was originally issued from the same hunting park as that of the wild boar cases. Altogether, the common point in all these French cases was the presence of deer in the parks/herds or close to cattle herds. This genotype has been circulating in France for the last 30 years, however such strains have been isolated sporadically which clearly shows that there is a lack of appropriate surveillance in animal parks. Given that primates are not maintenance hosts of *M. bovis*, the case described in the monkey—acting as a sentinel of the infection—could be the result of the under-detection of another wild ungulate case in the park. The low number of screening tests, such as the skin test, conducted on deer is the result of several factors, such as the difficulty to perform the test on these wild animals, the potential cross-reaction with environmental mycobacteria, or the inherent technical variability of the test [30]. 

Our data highlight the limit of classical genotyping techniques that can be overcome by the use of whole genome SNP analysis. Strains that shared the same genotype (spoligotype-VNTR) from French and foreign-born animals were genetically different and very distant.

The use of WGS revealed an unsuspected transmission network between hunting parks and zoos that needs to be further explored. These results reinforce the suspicion of an infectious deer-breeding network that could be at the origin of the bacillus dissemination. Indeed, this transmission could have occurred through inter-breeding animal exchanges. Our results raise awareness about the infectious risk of wild animals in captivity and the consequences of trading non-appropriately-tested animals. P22 ELISA, a currently available, highly specific and sensitive new test, could contribute to a more accurate diagnosis of TB in this species [30].

This study raises the problem of the zoonotic transmission of the disease, especially for cases in animal parks where close contact between free-roaming animals and the public is allowed. The fact that deer often develop generalized TB, and thus that they probably excrete the bacillus at significant rates, increases the risk to humans. This study therefore shows the need for the strengthened monitoring of these animals with regular ante-mortem tests (as for livestock) before allowing animal movement between parks.

## Figures and Tables

**Figure 1 microorganisms-07-00687-f001:**
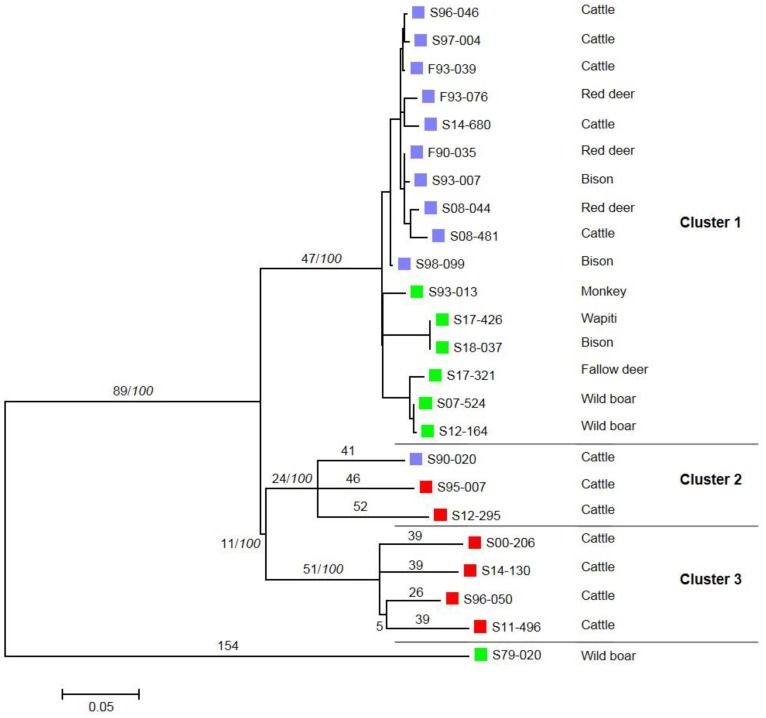
Molecular Phylogenetic tree of the 24 strains using the Maximum Likelihood method. The evolutionary history was inferred on MEGA by using the Maximum Likelihood method based on the General Time Reversible model on a final dataset of 770 SNP positions for 24 strains [1]. The number above each branch corresponds to the number of SNPs. Bootstrap values (%) based on 500 replications are indicated in italics after a slash if greater than 75%. A violet square represents strains isolated from French herds. A red square represents strains isolated from foreign-born animals. A green square represents strains isolated from park animals.

**Figure 2 microorganisms-07-00687-f002:**
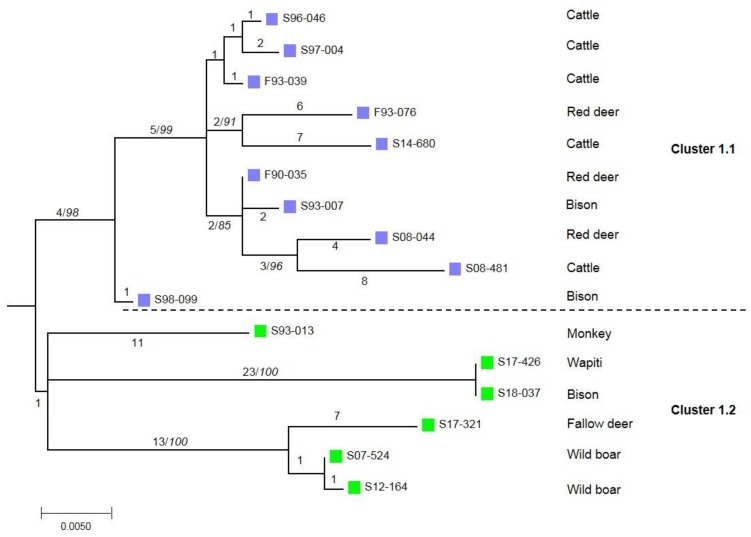
Zoomed-in view of the phylogenetic tree of cluster 1. The number above each branch corresponds to the number of SNPs. Bootstrap values (%) based on 500 replications are indicated in italics after a slash if greater than 75%. A violet square represents strains isolated from French herds. A green square represents strains isolated from park animals.

**Table 1 microorganisms-07-00687-t001:** Description of the 24 sequenced *M. bovis* strains.

		ID	Species	Year of Isolation	Context	Spoligotype	VNTR Profile	Accession Number
1	1979-02442	S79-020	Wild boar	1979	Park	GB54/SB0121	6 3 5 3 11 1 5 7	ERS3860435
2	1990-08827	F90-035	Red deer	1990	French herd	GB54/SB0121	6 4 5 3 11 2 5 7	ERS3860436
3	1990-09372	S90-020	Cattle	1990	French herd	F70/SB0295	6 4 5 3 11 2 5 7	ERS3860437
4	1993-04379	F93-039	Cattle	1993	French herd	GB54/SB0121	6 4 5 3 11 2 5 7	ERS3860438
5	1993-06259	S93-007	Bison	1993	French herd	GB54/SB0121	6 4 5 3 11 2 5 7	ERS3860439
6	1993-11575	F93-076	Red deer	1993	French herd	GB54/SB0121	6 4 5 3 11 2 5 7	ERS3860440
7	1993-11583	S93-013	Monkey	1993	Park	GB54/SB0121	6 4 5 3 11 2 5 7	ERS3860441
8	1995-00583	S95-007	Cattle	1995	Foreign-born animal	F070/SB0295	6 4 5 3 11 2 5 7	ERS3860442
9	1996-07003	S96-046	Cattle	1996	French herd	GB54/SB0121	6 4 5 3 11 2 5 7	ERS3860443
10	1996-08822	S96-050	Cattle	1996	Foreign-born animal	GB54/SB0121	6 4 5 3 11 2 5 7	ERS3860444
11	1997-00766	S97-004	Cattle	1997	French herd	GB54/SB0121	6 4 5 3 11 2 5 7	ERS3860445
12	1998-12646	S98-099	Bison	1998	French herd	GB54/SB0121	6 4 5 3 11 2 5 7	ERS3860446
13	00-06572	S00-206	Cattle	2000	Foreign-born animal	GB54/SB0121	6 4 5 3 11 2 5 7	ERS3860447
14	07-03151	S07-524	Wild boar	2007	Park	GB54/SB0121	6 4 5 3 11 2 5 7	ERS3860448
15	08-00202	S08-044	Red deer	2008	French herd	GB54/SB0121	6 4 5 3 11 2 5 7	ERS3860449
16	08-02744	S08-481	Cattle	2008	French herd	GB54/SB0121	6 4 5 3 11 2 5 7	ERS3860450
17	D11-02906	S11-496	Cattle	2011	Foreign-born animal	GB54/SB0121	6 4 5 3 11 2 5 7	ERS3860451
18	D12-01540	S12-164	Wild boar	2012	Park	GB54/SB0121	6 4 5 3 11 2 5 7	ERS3860452
19	D12-02012	S12-295	Cattle	2012	Foreign-born animal	F070/SB0295	6 4 5 3 11 2 5 7	ERS3860453
20	D14-00543	S14-130	Cattle	2014	Foreign-born animal	GB54/SB0121	6 4 5 3 11 2 5 7	ERS3860454
21	D14-02802	S14-680	Cattle	2014	French herd	GB54/SB0121	6 4 5 3 11 2 5 7	ERS3860455
22	D17-02544	S17-321	Fallow deer	2017	Park	GB54/SB0121	6 4 5 3 11 2 5 7	ERS3860456
23	D17-03607	S17-426	Wapiti	2017	Park	GB54/SB0121	6 4 5 3 11 2 5 7	ERS3860457
24	D18-00301	S18-037	Bison	2018	Park	GB54/SB0121	6 4 5 3 11 2 5 7	ERS3860458

**Table 2 microorganisms-07-00687-t002:** Estimates of Evolutionary Divergence between Sequences. The number of base differences between sequences are shown.

		F90-035	F93-039	S93-007	F93-076	S96-046	S97-004	S98-099	S08-044	S08-481	S14-680	S93-013	S07-524	S12-164	S17-321	S17-426	S18-037	S90-020	S95-007	S12-295	S96-050	S00-206	S11-496	S14-130
Cluster 1.1	F90-035																							
Cluster 1.1	F93-039	4																						
Cluster 1.1	S93-007	2	6																					
Cluster 1.1	F93-076	10	10	12																				
Cluster 1.1	S96-046	5	3	7	11																			
Cluster 1.1	S97-004	6	4	8	12	3																		
Cluster 1.1	S98-099	8	8	10	14	9	10																	
Cluster 1.1	S08-044	7	11	9	17	12	13	15																
Cluster 1.1	S08-481	11	15	13	21	16	17	19	12															
Cluster 1.1	S14-680	11	11	13	13	12	13	15	18	22														
Cluster 1.2	S93-013	23	23	25	29	24	25	17	30	34	30													
Cluster 1.2	S07-524	27	27	29	33	28	29	21	34	38	34	26												
Cluster 1.2	S12-164	28	28	30	34	29	30	22	35	39	35	27	1											
Cluster 1.2	S17-321	32	32	34	38	33	34	26	39	43	39	31	9	10										
Cluster 1.2	S17-426	35	35	37	41	36	37	29	42	46	42	34	38	39	43									
Cluster 1.2	S18-037	35	35	37	41	36	37	29	42	46	42	34	38	39	43	0								
Cluster 2	S90-020	134	134	136	140	135	136	128	141	145	141	135	139	140	144	147	147							
Cluster 2	S95-007	139	139	141	145	140	141	133	146	150	146	140	144	145	149	152	152	87						
Cluster 2	S12-295	145	145	147	151	146	147	139	152	156	152	146	150	151	155	158	158	93	98					
Cluster 3	S96-050	151	151	153	157	152	153	145	158	162	158	152	156	157	161	164	164	147	152	158				
Cluster 3	S00-206	159	159	161	165	160	161	153	166	170	166	160	164	165	169	172	172	155	160	166	70			
Cluster 3	S11-496	164	164	166	170	165	166	158	171	175	171	165	169	170	174	177	177	160	165	171	65	83		
Cluster 3	S14-130	159	159	161	165	160	161	153	166	170	166	160	164	165	169	172	172	155	160	166	68	78	81	
Outgroup	S79-020	301	301	303	307	302	303	295	308	312	308	302	306	307	311	314	314	319	324	330	336	344	349	344

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
