# Peer review of "Transmission Network of Deer-Borne Mycobacterium bovis Infection Revealed by a WGS Approach"

_microorganisms, 2019, doi:10.3390/microorganisms7120687_

Round 1

Reviewer 1 Report

Excellent use of a surveillance/monitoring program data analysis. Interesting information for parks and zoos given the infectious nature of TB. Potential for monitoring systems based on the tests used by authors.

Author Response

We are thankful to the Reviewer for this encouraging report.

Reviewer 2 Report

The authors analyzed SNPs of Mycobacterium bovis in livestock populations, wildlife, and animal parks. This data was used to prepare a phylogenetic tree to show that minimally 2 clear lines, one found in livestock and the other in zoos or hunting park animals.

The whole genome sequencing data is submitted to EBI, so the data is available to the public. The only data that are not shared that would be of interest to the community are the identity of the genes containing the 770 SNPs that were identified. Perhaps this information (or at least some additional information on the nature of the SNPs) could be added as a table in supporting information. Additionally, it would be of interest to know the identity of the 107 SNPs for cluster 1, etc.

The remaining data are provided as phylogenetic trees and support the conclusions provided by the authors.

Author Response

We are grateful to the Reviewer for the suggested improvements. As recommended, we have included a supplementary table providing information on the SNPs differentiating our strains.

Reviewer 3 Report

Approach done very interesting  and appreciable and in particular regarding WGS approoach respect deer borne and M. bovis, being M. bovis delicate and important germ in ONE health wildlife and domestic equilibrium in EU Community and not only in EU. 

Author Response

We are grateful to the Reviewer for this encouraging report.

Reviewer 4 Report

The research topic is interesting, but the sample size is small and there is no rigorous statistical analysis. 

Author Response

It is difficult to introduce any of the improvements the reviewer would have expected as they have not been accompanied by any argument to sustain them in his/her review report.

As discussed in our article, detection of tuberculosis in deer –captive or wild- is a great challenge given diagnostic technical constraints. Even more then in an officially bovine tuberculosis free country such as France, were these cases are scarce. From our point of view our collection is representative enough –different geographical sites and periods, different outbreaks- to sustain our findings. Besides, the reviewer alleges that no rigorous statistical analysis was performed, however, the bootstrap analyses employed for producing figures 1 and 2 include such analyses. Our results are thus statistically sustained.

Round 2

Reviewer 4 Report

all of my reviews have been well addressed.